# LEARNING FROM AGGREGATE-MASKED LABELS

## ABSTRACT

With the increasing concern over data privacy, more researchers are focusing on protecting sensitive labels using aggregate observations, such as similarity labels and label proportions. Unfortunately, these methods weaken the supervisory information of insensitive labels, thereby reducing the performance of existing classifiers. To address this issue, we propose a novel setting called Aggregate-Masked Labels, whose primary advantage lies in introducing augmented supervision to maintain partially full supervision and protecting sensitive labels. Specifically, for aggregate observations that contain sensitive labels, we use these sensitive labels as the aggregate-masked labels. In contrast, for aggregate observations without sensitive labels, we assign the ground-truth label to each instance, as shown in Figure 1. Moreover, we introduce the risk-consistent estimator that effectively leverages aggregate-masked labels to train a multi-class classifier. We further introduce stochastic label combinations to alleviate the high computational cost, effectively accelerating the training process. Experimental results on both real-world and benchmark datasets demonstrate that our method achieves state-of-the-art classification performance.

## 1 INTRODUCTION

As the demand for data privacy in machine learning applications grows (Wu et al., 2023; Wei et al., 2024), privacy-preserving methods in weakly supervised learning have become a critical area of research (Yao et al., 2023; Matsuo et al., 2024). The exposure of sensitive labels, particularly during the data annotation, can lead to significant risks, including privacy leakage, data misuse, and the potential for individuals to refuse providing ground-truth labels in sensitive contexts.

In recent years, researchers have made substantial efforts to alleviate this problem, including Partial-Label Learning (PLL) (Liang et al., 2025; Gong et al., 2025), Complementary Label Learning (CoLL) (Wang et al., 2024; Gao & Zhang, 2021; Gao et al., 2023), Multi-Positive and Unlabeled Learning (MPUL) (Xu et al., 2017) and Concealed Labels (CL) (Li et al., 2024). These methods assign each instance a set of candidate labels or leave it unlabeled. Moreover, aggregate observation methods represent another promising paradigm, including Similarity and Unlabeled Learning (SUL) (Wu et al., 2022; Nitayanont & Hochbaum, 2024), Similarity-Confidence Learning (SCL) (Zhang et al., 2024b; Barbany et al., 2024) and Label Proportion Learning (LPL) (Luo et al., 2024; Busa-Fekete et al., 2023). By replacing individual instance annotations with aggregate labels, these methods effectively protect sensitive labels during training and significantly reduce labeling costs.

However, we find that while aggregate observation methods are effective at protecting privacy, they often lack the full supervision from insensitive labels, potentially reducing the overall performance of the model. Then, we aim to propose a setting that protects sensitive labels and strengthens the supervision for insensitive labels. A straightforward but highly effective approach is to introduce augmented supervision for insensitive labels during the training process, which maintains partially full supervision for the classifier while protecting the sensitive labels.

Inspired by this finding, we propose a novel setting called Aggregate-Masked Labels (AML), which protects sensitive labels while maintaining partially full supervision from insensitive data. Specifically, AML uses an aggregate-masked label for aggregate observations containing sensitive labels to protect label privacy, while assigning ground-truth labels when all labels in the aggregate observation are insensitive. For example, as shown in Figure 1 (right), the aggregate observation includes a sen-

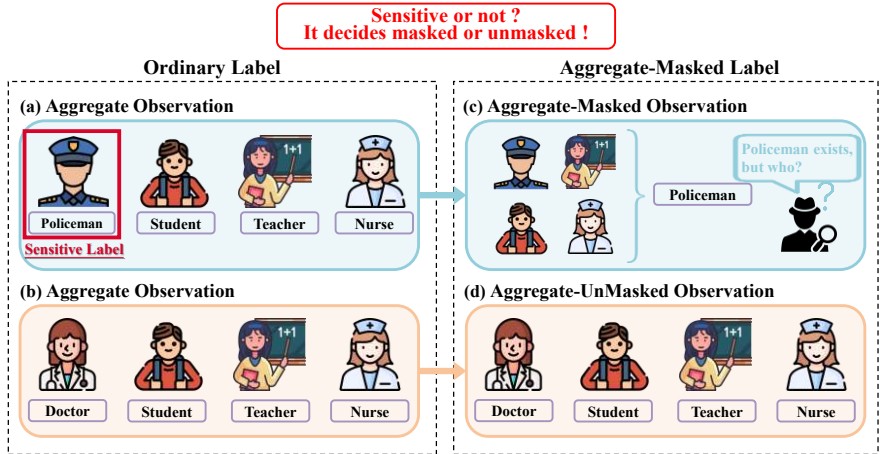

Figure 1: Illustration of the comparison between the ordinary labeling setting and the AML setting. In many cases, certain labels, such as "Policeman", are considered sensitive. In the ordinary labeling setting (left), all instances are annotated with ground-truth labels, including sensitive ones, which may lead to privacy leakage. In the AML setting (right), if the aggregate observation contains sensitive labels, the ground-truth labels of instances are not exposed. Instead, the set of sensitive labels within the aggregate observation is provided as AML (e.g., "Policeman"), ensuring the protection of privacy. For insensitive labels, such as "Doctor", "Student", "Teacher" or "Nurse" are retained.

sitive label such as "Policeman", the entire aggregate observation is assigned an aggregate-masked label, which protects private data, i.e., adversaries cannot identify the specific instance associated with the sensitive label. In contrast, if all labels within an aggregate observation are insensitive (e.g., "Doctor", "Nurse", "Student", "Teacher"), the ground-truth labels are maintained. The primary advantage of aggregate-masked labels is that they provide aggregate labels without exposing sensitive labels and are often easier to obtain in practice.

In this paper, we propose a risk-consistent estimator to learn from these aggregate observations. The estimator utilizes full supervision from insensitive labels and aggregate-masked labels from aggregate observations. In addition, we introduce stochastic label combinations to alleviate the high computational cost, effectively accelerating the training process. Experiments on real-world and benchmark datasets demonstrate that our approach achieves state-of-the-art classification performance without disclosing private data. Our main contributions are as follows:

- We propose a novel setting, i.e., learning from aggregate-masked labels, which masks sensitive labels while maintaining partially full supervision for insensitive labels.

- We propose a risk-consistent estimator that effectively leverages aggregate-masked labels to train a multi-class classifier, which classify precisely instances from insensitive and sensitive labels.

- We propose a stochastic label combination method that reduces the high computational cost of enumerating all labels under the AML setting. Combined with probability estimation, our approach better exploit the aggregate-masked labels, effectively accelerating training.

## 2 METHODOLOGY

In this section, we introduce the problem setting for AML and propose the corresponding method, including the risk-consistent estimator and stochastic aggregate-masked observation risk.

### 2.1 PROBLEM SETUP

**Sensitive labels.** Let $\mathcal{X} \subset \mathbb{R}^d$ denotes the $d$-dimensional feature space, and $\mathcal{Y} \triangleq \mathcal{I} \bigcup \mathcal{S} = \{1, \ldots, r, s_1, \ldots, s_t\}$ denotes the label space, where $\mathcal{I} = \{1, \ldots, r\}$ denotes the set of $r$ insensitive labels, and $\mathcal{S} = \{s_1, \ldots, s_t\}$ denotes the set of $t$ sensitive labels. In real scenarios, distinguishing

labels between sensitive (e.g., "Policeman") and insensitive (e.g., "Student") is essential for privacy label annotation. Sensitive labels may expose attributes associated with personal privacy, whereas insensitive labels maintain partially full supervision without disclosing privacy.

**Aggregate observations.** Let $X = \{x_1, \ldots, x_m\}$ denotes the aggregate observation sampled from distribution $p(X)$, and let $Y = \{y_1, \ldots, y_m\} \in \mathcal{Y}^m$ denotes the ordinary supervised labels of the aggregate observations, where $\mathcal{Y}^m$ denotes the product set of $\mathcal{Y}$. The goal of this setting is to learn a classifier $f : \mathcal{X} \to \mathcal{Y}$ that not only recognizes insensitive labels, but also accurately predicts sensitive labels by minimizing the expected aggregate observations ordinary supervised risk as follows:

$$R(f) = \mathbb{E}_{(X,Y) \sim p(X,Y)} \sum_{i=1}^{m} \mathcal{L}\left[f(x_i), y_i\right]$$

$$= \mathbb{E}_{X \sim p(X)} \sum_{Y \in \mathcal{Y}^m} p(Y \mid X) \sum_{i=1}^{m} \mathcal{L}\left[f(x_i), y_i\right] \tag{1}$$

where $p(X, Y)$ and $p(Y \mid X)$ denote the joint and conditional distributions of the aggregate observation data, and $\mathcal{L}\left[f(x_i), y_i\right]$ denotes the multi-class loss function.

**Aggregate-Masked Labels.** In our setting, we collect aggregate observation $X = \{x_1, \ldots, x_m\}$ and provide the Aggregate-Masked Labels (AML) $Y_A \in \mathcal{Y}_A$ to form an AML observation $A_i = (\{x_{i,1}, \ldots, x_{i,j}, \ldots, x_{i,m}\}, Y_{A,i})$, where $\mathcal{Y}_A$ denotes the label space of AML and $x_{i,j}$ denotes the $j_{th}$ instance within the $i_{th}$ AML observation. In this paper, as shown in Figure.1, we called AML observation $A_i$ as Aggregate-Masked Observation (AMO), when it contains at least one sensitive label, i.e., $\mathcal{S} \cap \{y_{i,1}, \ldots, y_{i,j} \ldots, y_{i,m}\} \neq \varnothing$, where $y_{i,j}$ is $x_{i,j}$ corresponding ground-truth label. In contrast, we called it as Aggregate-UnMasked Observation (AUMO), i.e., $\forall j, y_{i,j} \in \mathcal{I}$.

Let $\mathcal{Y}_U = \{1, \ldots, r\}^m$ denotes the label space of AUMO, $\mathcal{Y}_M = \mathcal{P}(\mathcal{S})/\varnothing$ denotes the label space of AMO, and $\mathcal{Y}_A = \mathcal{Y}_U \bigcup \mathcal{Y}_M$, where $\mathcal{P}(\mathcal{S})$ denotes the set of all subsets of sensitive labels. Then, the $i_{th}$ AML $Y_{A,i}$ can be defined as follows:

$$Y_{A,i} = \begin{cases} Y_{U,i} & \text{if} \quad \forall j, y_{i,j} \in \mathcal{I} \quad \text{(AUMO)}, \\ Y_{M,i} & \text{else} \quad \text{(AMO)} \end{cases} \tag{2}$$

where $Y_{U,i} = \{y_{i,1}, \ldots, y_{i,m}\} \in \mathcal{Y}_U$, $Y_{M,i} = \mathcal{S} \cap \{y_{i,1}, \ldots, y_{i,m}\}$, and satisfies $Y_{M,i} \subseteq \mathcal{Y}_M$. Let $\mathcal{D} = \{A_i\}_i^N$ denotes the AML dataset, where $N$ is the total number of AML observation. Notably, to protect sensitive labels, we have established a constraint on the number of samples $m$. Specifically, within each AML observation, $m$ is required to meet the condition $m \geq t + 1$.

**Motivation.** By introducing augmented supervision, AML not only protects sensitive labels but also maintains partially full supervision for insensitive labels, which enables the model to leverage more supervision during training and thereby achieve better classification performance. For example, the aggregate $A_i$ contains four samples in Figure 1 (a), among which one sample belongs to the sensitive label "Policeman". Then, the aggregate is labeled as $Y_{A,i} = \{Policeman\}$, which protects the sensitive label in the data collection. In contrast, for aggregated observations that do not contain sensitive labels in Figure 1 (b), the ground-truth label of each sample is preserved, thereby significantly enriching the supervisory information.

To achieve a more formal comprehension of the characteristics of AML, we present the following assumptions. Assumption 1 describes the conditional distribution relationship between ordinary labels and AML, thereby enabling the construction of a risk-consistent estimator.

**Assumption 1** (AML Assumption). The conditional distribution of AML are under the assumption as follows:

$$P(\forall i, y_i \in \mathcal{I} \mid Y_A \in \mathcal{Y}_U, X) = 1 \tag{3}$$

This assumption indicates that for any aggregate observations, if its AML $Y_A$ is belong to $\mathcal{Y}_U$, then the ordinary labels of their instances must be the insensitive label.

## 2.2 RISK-CONSISTENT ESTIMATOR

In this section, we propose a risk-consistent estimator for the classification task under AML setting, aiming to effectively learn from Aggregate-Masked Observations.

As shown in Eq. (1), computing the expected ordinary supervised risk requires access to the ground-truth labels $Y = \{y_1, \ldots, y_m\}$. However, for AML of AMO ($Y_A \in \mathcal{Y}_M$), these labels are unavailable due to the presence of sensitive labels. To address this, we introduce a risk-consistent estimator that approximates the unknown expectation over $Y$ using the predicted label distributions. Specifically, we evaluate the loss over all possible label assignments within AMO, and subtract the contribution from labels in $\mathcal{Y}_U$, maintaining only the influence of labels in $\mathcal{Y}_M$. This decomposition is formally characterized in Lemma 2, which enables the computation of the AMO risk using only AML and model predictions, without requiring ground-truth supervision.

**Lemma 2.** (AMO risk) Under the AML assumption, for multi-class classifier $f$, we derive the expected risk for aggregate observations in the AMO (i.e., $Y_A \in \mathcal{Y}_M$) as follows:

$$
\begin{aligned}
R_{AMO}(f) = {}& \mathbb{E}_{X \sim p(X)} \sum_{Y_A \in \mathcal{Y}_M} \sum_{Y \in \mathcal{Y}^m} p(Y, Y_A \mid X) \sum_{i=1}^{m} \mathcal{L}\left[f(x_i), y_i\right] \\
= {}& \mathbb{E}_{(X, Y_A) \sim p(X, Y_A \in \mathcal{Y}_M)} \left[ \sum_{k_1 \in \mathcal{Y}_U \bigcup Y_A} \cdots \sum_{k_m \in \mathcal{Y}_U \bigcup Y_A} \hat{L}(X, K) \right. \\
& \left. - \sum_{k_1 \in \mathcal{Y}_U} \cdots \sum_{k_m \in \mathcal{Y}_U} \hat{L}(X, K) \right]
\end{aligned}
\tag{4}
$$

where $\hat{L}(X, K)$ denotes the product of the predicted probabilities for $X$ under label assignment $K = \{k_1, \ldots, k_m\}$, multiplied by the sum of the label losses $\mathcal{L}\left[f(x_i), k_i\right]$ for all instance. Formally, it can be defined as:

$$
\hat{L}(X, K) = \prod_{j=1}^{m} P(y_j = k_j \mid x_j) \sum_{i=1}^{m} \mathcal{L}\left[f(x_i), k_i\right]
\tag{5}
$$

Here, $P(y_j = k_j \mid x_j)$ represents the prediction probability of the model that instance $x_j$ assigned to class $k_j$, and $\mathcal{L}\left[f(x_i), y_i\right]$ is the loss about predicted label $y_i$ for instance $x_i$. The proof of Lemma 2 is provided in the Appendix B.1.

Therefore, the total expected risk can be decomposed into two parts. Specifically, for the risk corresponding to the AUMO, we can compute it directly using the model predictions for each instance. However, for AMO, we adjust the risk calculation based on Eq. (4), leading to the total classification risk under the AML assumption as follows.

**Theorem 3.** Under the AML assumption, the classification risk $R(f)$ in Eq. (1) can be equivalently expressed as follows:

$$
\begin{aligned}
R_{AML}(f) = {}& \mathbb{E}_{X, Y_A \sim p(X, Y_A \in \mathcal{Y}_U)} \sum_{i=1}^{m} \mathcal{L}\left[f(x_i), Y_{A_i}\right] \\
& + \mathbb{E}_{X, Y_A \sim p(X, Y_A \in \mathcal{Y}_M)} \left[ \sum_{k_1 \in \mathcal{Y}_U \bigcup Y_A} \cdots \sum_{k_m \in \mathcal{Y}_U \bigcup Y_A} \hat{L}(X, K) \right. \\
& \left. - \sum_{k_1 \in \mathcal{Y}_U} \cdots \sum_{k_m \in \mathcal{Y}_U} \hat{L}(X, K) \right]
\end{aligned}
\tag{6}
$$

The proof of Theorem 3 is provided in Appendix B.2. It is worth noting that this risk formulation characterizes the relationship between learning under the AML setting and ordinary supervised learning, and demonstrates the influence of AML on model training. The first term corresponds to the AUMO setting, where each instance $x_i$ is supervised by $Y_{U,i}$, and the loss can be directly computed. The second term corresponds to the AMO setting, where the AML $Y_{M,i}$ is unknown.

**Empirical risk.** Since the training dataset $\mathcal{D}$ is sampled independently from the $p(X, Y)$, the empirical risk estimator can be naively approximated as:

$$\widehat{R}_{AML}(f) = \frac{1}{N} \sum_{j=1}^{N} \sum_{i=1}^{m} \mathcal{L}\left[f(x_{j,i}), Y_{A,j,i}\right]$$

$$+ \frac{1}{N} \sum_{j=1}^{N} \left[ \sum_{k_1 \in \mathcal{Y}_U \bigcup Y_{A,j}} \cdots \sum_{k_m \in \mathcal{Y}_U \bigcup Y_{A,j}} \hat{L}(X_j, K) \right.$$

$$\left. - \sum_{k_1 \in \mathcal{Y}_U} \cdots \sum_{k_m \in \mathcal{Y}_U} \hat{L}(X_j, K) \right] \tag{7}$$

where $x_{j,i}$ and $Y_{A,j,i}$ respectively denote the $i_{th}$ instance and its label in the $j_{th}$ aggregation observations, as obtained from AUMO.

## 2.3 STOCHASTIC AMO RISK

Although the formulation in Eq. (6) is effective, it sums over all possible combinations of labels from $Y_A$. When the aggregate observations size $m$ and labels space $\mathcal{Y}$ are large, the required computations grow exponentially, reaching up to $|\mathcal{Y}|^m$ during training. Then, we introduce a stochastic sampling strategy to reduce the computational cost of enumerating all label in the $R_{AMO}(f)$.

Specifically, let $\alpha_K$ be a binary variable that takes a value of either 0 or 1. We uniformly sample a fixed number of elements from all possible label combinations, and assign $\alpha_K$ a value of 1 for the selected elements; otherwise, $\alpha_K$ is assigned a value of 0. Then, we propose a method called stochastic AMO risk for accelerating training, defined as follows:

$$R_{SAMO}(f) = \mathbb{E}_{X,Y_A \sim p(X,Y_A \in \mathcal{Y}_M)} \left[ \sum_{k_1 \in \mathcal{Y}_U \bigcup Y_A} \cdots \sum_{k_m \in \mathcal{Y}_U \bigcup Y_A} \alpha_K \, \hat{L}(X, K) \right.$$

$$\left. - \sum_{k_1 \in \mathcal{Y}_U} \cdots \sum_{k_m \in \mathcal{Y}_U} \alpha_K \, \hat{L}(X, K) \right] \tag{8}$$

By leveraging this risk, we can control the number of label combinations, thereby achieving the objective of reducing computational costs.

## 2.4 PRACTICAL IMPLEMENTATION

**Probability estimator.** In this section, we introduce CLIP (Contrastive Language-Image Pretraining) to estimate label probabilities in AML settings. Given sample $x_i$, CLIP generates a probability distribution over a set of class labels $y_i$. This distribution can be formally represented as $P(y_i = k_i|x_i)$, where $k_i$ is the predicted class for the $i_{th}$ instance. In the context of AMO, where only AML are available and ground-truth labels are unavailable, these predictions serve as approximations of labels. We incorporate these probability estimates into the risk-consistent formulation, which enhances the model's performance under the AML setting and demonstrates its effectiveness.

**Loss functions.** Designing effective loss functions is essential for AML. To address this, we employ the squared loss under the One-Versus-Rest (OVR) framework, which encourages positive alignment between predictions and their corresponding labels. The OVR strategy offers theoretical guarantees and has demonstrated strong empirical performance in multi-class supervised learning scenarios, making it a practical and reliable choice in our setting.

**Model.** We adopt the CLIP ViT-L/14 visual encoder as the backbone of our model, leveraging its strong semantic alignment capabilities between visual and textual modalities. To preserve the rich representations learned during pretraining, the image encoder is kept frozen throughout the training process. The high-dimensional embeddings produced by CLIP are subsequently passed through multiple linear classification layers to predict labels. This design improves the efficiency of feature utilization while reducing the risk of overfitting, enabling robust learning under the masked label.

Table 1: Classification accuracy ( $mean \pm std$ ) of each algorithm on CIFAR-10, Caltech-101 and DTD. $R$ denotes a random choice. The best result in each setting is highlighted in **bold**, and the second-best is underlined.

| Dataset | Method | Sensitive Labels Set $\mathcal{S}$ | | | | | |
|---|---|---|---|---|---|---|---|
| | | {0} | {1} | {0, 1} | {1, 2} | {R, R} | {R, R} |
| CIFAR-10 | ESA (Li et al., 2025b) | 87.30±0.09 | 85.82±0.23 | 79.02±0.19 | 79.19±0.26 | 85.66±0.33 | 70.19±0.15 |
| | CoMPU (Zhou et al., 2022) | 85.86±0.07 | 87.66±0.04 | 74.49±0.18 | 78.63±0.04 | 87.04±0.01 | 69.93±0.12 |
| | NMPU (Shu et al., 2020) | 86.38±0.05 | 84.55±0.52 | 72.66±0.63 | 78.02±0.33 | 84.15±0.21 | 69.07±0.20 |
| | PAPI (Xia et al., 2023) | 82.15±0.68 | 79.41±1.69 | 81.04±0.19 | 73.51±3.52 | 77.88±2.13 | 79.76±3.06 |
| | SPMI (Liu et al., 2024) | 51.22±0.03 | 50.88±0.12 | 49.27±0.23 | 47.10±0.32 | 51.72±0.02 | 50.47±0.09 |
| | DIRK (Wu et al., 2024) | 72.40±1.10 | 69.07±0.41 | 74.40±0.74 | 67.51±0.74 | 69.28±0.78 | 67.51±0.74 |
| | L$^2$P-AHIL (Ma et al., 2025) | 62.81±0.14 | 62.51±0.39 | 63.27±0.36 | 63.30±0.16 | 63.22±0.33 | 63.37±0.39 |
| | **AML** | **89.95±0.04** | **89.96±0.12** | **89.36±0.23** | **88.31±0.10** | **88.10±0.83** | **88.82±0.13** |
| Caltech-101 | ESA (Li et al., 2025b) | 54.14±2.19 | 53.95±0.54 | 59.98±0.03 | 61.49±1.47 | 56.12±0.31 | 63.57±0.39 |
| | CoMPU (Zhou et al., 2022) | 38.11±0.82 | 43.40±5.57 | 44.67±0.62 | 43.29±0.05 | 47.77±0.25 | 39.08±1.26 |
| | NMPU (Shu et al., 2020) | 50.07±0.41 | 52.88±0.51 | 43.20±0.85 | 48.94±2.01 | 53.41±1.42 | 51.56±1.08 |
| | PAPI (Xia et al., 2023) | 72.99±0.27 | 72.63±0.55 | 69.91±1.78 | 69.61±0.85 | 72.45±0.62 | 72.50±0.59 |
| | SPMI (Liu et al., 2024) | 67.43±0.36 | 66.79±0.60 | 66.15±0.55 | 66.19±1.13 | 67.91±0.73 | 68.24±0.69 |
| | DIRK (Wu et al., 2024) | 62.55±0.35 | 65.10±1.09 | 63.02±0.59 | 62.37±0.32 | 64.78±0.68 | 64.31±1.56 |
| | L$^2$P-AHIL (Ma et al., 2025) | 46.40±1.09 | 46.97±0.54 | 48.49±0.74 | 48.24±0.54 | 46.89±0.77 | 47.25±0.71 |
| | **AML** | **81.99±0.00** | **73.39±0.21** | **72.78±0.00** | **73.02±0.31** | **81.59±0.05** | **81.14±0.07** |
| DTD | ESA (Li et al., 2025b) | 55.41±0.88 | 56.64±1.75 | 67.60±1.56 | 67.64±0.25 | 70.07±0.56 | 69.01±0.43 |
| | CoMPU (Zhou et al., 2022) | 44.33±0.75 | 36.70±0.13 | 51.99±0.56 | 49.33±1.31 | 55.18±2.19 | 58.55±3.32 |
| | NMPU (Shu et al., 2020) | 62.23±0.37 | 62.05±0.50 | 62.72±1.44 | 60.15±0.94 | 63.60±1.44 | 58.68±0.50 |
| | PAPI (Xia et al., 2023) | 50.96±0.72 | 50.93±0.39 | 50.39±0.38 | 50.32±0.52 | 50.24±1.01 | 50.17±0.77 |
| | SPMI (Liu et al., 2024) | 47.60±0.75 | 46.97±1.69 | 45.80±1.21 | 46.79±0.52 | 46.52±0.31 | 47.83±0.69 |
| | DIRK (Wu et al., 2024) | 46.04±0.50 | 45.60±1.17 | 45.57±0.55 | 46.72±0.67 | 47.20±0.19 | 38.54±0.71 |
| | L$^2$P-AHIL (Ma et al., 2025) | 17.33±0.56 | 16.86±0.54 | 16.08±0.80 | 16.19±0.08 | 17.85±0.35 | 17.36±0.43 |
| | **AML** | **80.90±0.06** | **81.80±0.22** | **80.27±0.44** | **79.69±0.38** | **78.07±0.13** | **80.96±0.36** |

## 3 EXPERIMENTS

### 3.1 EXPERIMENTAL SETUP

**Datasets.** To comprehensively evaluate the effectiveness of our method, we conduct experiments on several standard image classification benchmarks as well as two real-world datasets. The benchmark datasets include CIFAR10 (Krizhevsky et al., 2009), CIFAR100 (Krizhevsky et al., 2009), Caltech-101 (Fei-Fei et al., 2007), and DTD (Cimpoi et al., 2014). The real-world datasets include AMLM and AMLS. The details of the datasets are provided in the Appendix C.

Table 2: Classification accuracy ($mean \pm std$) of each algorithm on CIFAR-100. $R$ denotes a random choice. The best result is in **bold**, and the second-best is underlined.

| Method | Sensitive Labels Set $\mathcal{S}$ | | |
|---|---|---|---|
| | {0} | {0, 1} | {R, R} |
| ESA (Li et al., 2025b) | 51.26±0.29 | 57.90±0.22 | 57.98±0.21 |
| CoMPU (Zhou et al., 2022) | 33.86±1.29 | 31.54±2.33 | 28.53±0.75 |
| NMPU (Shu et al., 2020) | 28.06±0.13 | 26.51±0.50 | 24.70±1.13 |
| PAPI (Xia et al., 2023) | 28.59±1.00 | 28.22±0.56 | 29.13±0.74 |
| SPMI (Liu et al., 2024) | 19.89±0.66 | 19.94±0.74 | 19.44±0.17 |
| DIRK (Wu et al., 2024) | 9.83±0.62 | 9.84±0.17 | 8.73±1.17 |
| L$^2$P-AHIL (Ma et al., 2025) | 36.58±0.20 | 35.77±0.16 | 35.39±0.40 |
| **AML** | **68.31±0.12** | **67.43±0.08** | **66.63±0.25** |

**Compared approaches.** We compare our proposed method with representative approaches from three related methods: Multi-Positive and Unlabeled Learning (ESA (Li et al., 2025b), CoMPU (Zhou et al., 2022), NMPU (Shu et al., 2020)), Partial-Label Learning (PAPI (Xia et al., 2023), SPMI (Liu et al., 2024), DIRK (Wu et al., 2024)) and Label Proportion Learning (L$^2$P-AHIL (Ma et al., 2025)). Specifically, for ESA, CoMPU and NMPU, we treat insensitive labels as positive and sensitive labels as negative. To enable a fair comparison, we assume class priors are known during training. For PAPI, SPMI and DIRK, we construct candidate label matrices according to their respective assumptions. Specifically, instances in the AUMO setting are assigned a positive label, while those in the AMO setting are labeled AML. For L$^2$P-AHIL, we apply proportion-level supervision to groups containing sensitive labels, while groups without sensitive labels are trained with instance-level supervision.

**Implementation details.** We implement all methods using PyTorch and conduct training under consistent computational settings on a single NVIDIA RTX 4090D GPU. The models are optimized using the AdamW optimizer with an initial learning rate selected from $\{1e^{-4}, 1e^{-3}, 1e^{-2}, 1e^{-1}\}$, and a weight decay of $\{1e^{-3}, 1e^{-4}\}$. In addition, each method is run three times, and we report the mean and standard deviation of the results.

Table 3: Classification accuracy ( $mean \pm std$ ) of each algorithm on AMLM and AMLS. $R$ denotes a random choice. The best result is highlighted in **bold**, and the second-best is underlined.

| Dataset | Method | Sensitive Labels Set $\mathcal{S}$ | | | | | |
|---------|--------|-------|-------|--------|--------|----------|----------|
| | | {0} | {1} | {0, 1} | {1, 2} | {R, R} | {R, R} |
| AMLM | ESA (Li et al., 2025b) | 87.34±1.79 | 85.44±0.00 | 80.06±2.24 | 79.43±1.34 | 84.49±2.24 | 75.32±1.79 |
| | CoMPU (Zhou et al., 2022) | 82.28±3.58 | 87.02±1.34 | 77.53±0.45 | 75.63±0.45 | 72.78±0.00 | 71.52±2.68 |
| | NMPU (Shu et al., 2020) | 71.52±2.68 | 85.12±0.44 | 69.30±2.24 | 73.10±0.44 | 78.79±1.34 | 61.07±1.34 |
| | PAPI (Xia et al., 2023) | 54.24±1.28 | 51.55±3.83 | 52.38±2.88 | 50.93±5.15 | 57.14±1.01 | 48.45±1.52 |
| | SPMI (Liu et al., 2024) | 58.39±2.21 | 55.28±0.51 | 51.14±2.60 | 55.69±0.29 | 55.90±1.34 | 47.62±0.59 |
| | DIRK (Wu et al., 2024) | 64.39±1.46 | 68.26±1.37 | 60.46±1.46 | 70.45±0.22 | 70.45±0.22 | 47.67±2.46 |
| | L$^2$P-AHIL (Ma et al., 2025) | 49.27±4.05 | 49.83±5.16 | 47.27±0.96 | 46.29±1.69 | 49.31±1.02 | 46.37±1.47 |
| | **AML** | **88.82±0.88** | **90.06±0.62** | **86.02±1.32** | **90.06±1.24** | **86.95±1.07** | **85.40±0.44** |
| AMLS | ESA (Li et al., 2025b) | 79.51±1.16 | 74.32±1.54 | 73.22±1.54 | 65.84±1.15 | 51.09±2.70 | 75.13±1.15 |
| | CoMPU (Zhou et al., 2022) | 80.05±0.38 | 80.87±0.77 | 72.95±1.16 | 65.57±0.77 | 84.15±1.54 | 67.21±1.54 |
| | NMPU (Shu et al., 2020) | 78.41±0.38 | 80.32±0.00 | 67.75±0.77 | 57.10±1.15 | 50.00±1.16 | 59.28±0.38 |
| | PAPI (Xia et al., 2023) | 45.59±0.25 | 39.46±0.44 | 43.24±1.53 | 47.57±1.17 | 41.80±1.35 | 43.24±3.50 |
| | SPMI (Liu et al., 2024) | 56.04±1.27 | 53.51±1.17 | 58.20±1.67 | 56.58±1.35 | 55.14±1.17 | 54.05±0.44 |
| | DIRK (Wu et al., 2024) | 39.91±1.16 | 37.51±0.23 | 41.67±1.76 | 46.71±1.02 | 39.69±0.06 | 38.66±1.37 |
| | L$^2$P-AHIL (Ma et al., 2025) | 52.32±2.78 | 54.52±3.63 | 48.00±4.66 | 58.13±1.40 | 57.56±2.11 | 50.41±5.00 |
| | **AML** | **85.41±1.53** | **88.83±1.56** | **82.43±1.53** | **77.84±1.87** | **89.83±1.43** | **81.62±0.94** |

Table 4: Classification accuracy ( $mean \pm std$ ) of different number of sensitive labels on CIFAR-10, Caltech-101, DTD, AMLM and AMLS. We denote the number of sensitive labels as $t$.

| Number | CIFAR-10 | Caltech-101 | DTD | AMLM | AMLS |
|--------|----------|-------------|-----|------|------|
| $t = 1$ | 89.95±0.04 | 82.03±0.05 | 80.90±0.06 | 88.82±0.88 | 85.41±1.53 |
| $t = 2$ | 89.36±0.23 | 72.50±0.38 | 80.27±0.44 | 86.02±1.32 | 82.43±0.38 |
| $t = 3$ | 77.74±0.17 | 72.34±0.67 | 77.88±0.06 | 82.92±2.20 | 70.81±4.59 |
| $t = 4$ | 68.70±0.27 | 71.73±0.03 | 75.93±0.18 | 71.42±3.51 | 47.02±2.29 |
| $t = 5$ | 51.81±0.61 | 66.91±0.00 | 73.31±0.12 | 64.59±0.00 | 33.51±1.58 |

## 3.2 RESULTS ON BENCHMARK DATA

We conduct experiments on benchmark datasets and training label combinations are stochastically selected based on the size of the sensitive label set $\mathcal{S}$. Specifically, for $|\mathcal{S}| = 1$, all samples are included (100%). For $|\mathcal{S}| = 2$, only the top 20% of samples are selected. This stochastic sampling strategy effectively reduces the training cost for larger $|\mathcal{S}|$. As shown in Table 1 and Table 2, the proposed method achieves the best performance in all datasets. Notably, DIRK performs poorly on CIFAR-100 as its rectification mechanism fails to recover supervision with increasing classes. These results demonstrate the effectiveness and robustness of our method.

## 3.3 RESULTS ON REAL-WORLD DATA

To further evaluate the effectiveness of our method in real-world scenarios, we conduct experiments on two datasets: AMLM and AMLS. The experimental setting, including the stochastic sampling based on the size of the sensitive label set $\mathcal{S}$, follows the same setting used in the benchmark datasets. As shown in Table 3, the proposed method achieves the best overall performance in both datasets. Specifically, our approach consistently outperforms state-of-the-art weakly supervised methods such as ESA, CoMPU, and NMPU. On the other hand, the inferior performance of PAPI, SPMI, and DIRK can be attributed to the reduced supervision they provide in the privacy-preserving setting. These results clearly validate the superiority of our method compared to all baselines, highlighting its robustness and strong adaptability to privacy classification tasks.

## 3.4 EFFECT OF THE NUMBER OF SENSITIVE LABELS

To further validate the robustness of our method under AML, we conduct an ablation study on the five dataset by varying the number of sensitive labels, i.e., $t \in \{1, 2, 3, 4, 5\}$. As shown in Table 4, the classification performance degrades with the increase in the number of sensitive labels, which can be attributed to decreasing supervision. Our method maintains high accuracy and stability even when many labels are masked, demonstrating its effectiveness in protecting sensitive labels.

Table 5: Classification accuracy ( $mean \pm std$ ) of different aggregate observations size on CIFAR-10. Given $t$ sensitive labels, we evaluate model performance with the number of instances $m$.

| Dataset | $\mathcal{S}$ | $m = t+1$ | $m = t+2$ | $m = t+3$ | $m = t+4$ | $m = t+5$ |
|---|---|---|---|---|---|---|
| | $\{0\}$ | 89.95±0.04 | 89.35±0.14 | 80.91±0.11 | 80.76±0.07 | 80.61±0.07 |
| | $\{1\}$ | 89.96±0.12 | 89.73±0.21 | 80.87±0.09 | 80.84±0.12 | 80.58±0.05 |
| CIFAR-10 | $\{0,1\}$ | 89.36±0.23 | 81.77±4.39 | 71.40±0.07 | 71.13±0.03 | 70.86±0.02 |
| | $\{1,2\}$ | 88.31±0.10 | 78.28±0.71 | 71.99±0.16 | 71.63±0.00 | 71.23±0.09 |
| | $\{0,1,2\}$ | 77.74±0.17 | 62.34±0.00 | 62.12±0.11 | 61.47±0.07 | 60.99±0.00 |

Table 6: Classification accuracy ( $mean \pm std$ ) of different CLIP-based Text Prompts on CIFAR-10. $R$ denotes a random choice.

| Prompt | Sensitive Labels Set $\mathcal{S}$ | | | | | |
|---|---|---|---|---|---|---|
| | $\{0\}$ | $\{1\}$ | $\{0,1\}$ | $\{1,2\}$ | $\{R,R\}$ | $\{R,R\}$ |
| Default | 89.95±0.04 | 89.96±0.12 | 89.36±0.23 | 88.31±0.10 | 88.82±0.13 | 89.83±0.00 |
| GPT-4o | 89.89±0.07 | 90.11±0.05 | 89.16±0.19 | 88.46±0.15 | 88.74±0.19 | 88.78±0.17 |
| ERNIE-Bot | 89.86±0.07 | 89.92±0.09 | 88.27±0.36 | 87.44±0.39 | 89.59±0.11 | 89.75±0.10 |
| DeepSeek | 90.10±0.07 | 90.11±0.16 | 89.30±0.16 | 89.28±0.27 | 89.06±0.17 | 89.16±0.02 |

### 3.5 EFFECT OF AGGREGATE OBSERVATIONS SIZE

To evaluate the sensitivity of our method for the aggregate observations, we conduct experiments by varying the size of aggregate observations. Specifically, as shown in Table 5, the original setting assigns $m = t + 1$ labels to each aggregate observation, where $t$ denotes the number of sensitive labels. Then, we increase the size of aggregate observations by adding two to five samples. From the table, we can observe that as the size of the aggregate observations increases, the classifier's performance slightly decreases. This trend is expected, since the larger observation size reduces the amount of supervision available. However, the model's overall performance remains competitive, since augmented supervision introduces partially full supervision to offset the reduced information.

### 3.6 EFFECT OF STOCHASTIC AMO RISK

In this section, we further investigate the impact of varying stochastic label combinations. Specifically, we analyze the convergence behavior under two sensitive label sets, $\mathcal{S} = \{0\}$ and $\mathcal{S} = \{0, 1\}$, using 10%, 20%, 50% and 100% of the stochastically selected label combinations. As shown in Figure 2, the model converges to comparable accuracy even when trained with significantly fewer label combinations (e.g., 20% or 50%). However, in the $\mathcal{S} = \{0, 1\}$, using only 10% of the combinations results in a slight performance drop. We believe that a small number of combinations in a larger combination space leads to insufficient supervision, resulting in slightly lower accuracy. These results demonstrate the robustness and efficiency of our stochastic label combinations.

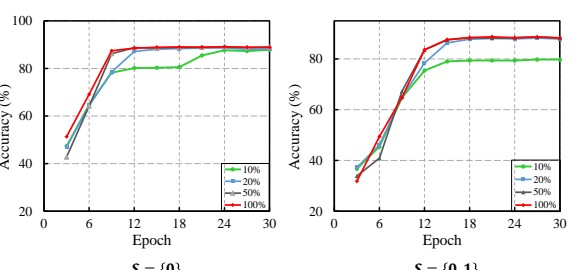

Figure 2: Classification accuracy under different stochastic label combinations and model convergence across sampling rates $(10\%, 20\%, 50\%, 100\%)$ with sensitive label sets $\mathcal{S} = \{0\}$ and $\mathcal{S} = \{0, 1\}$.

### 3.7 EFFECT OF CLIP-BASED TEXT PROMPTS

To further investigate the impact of CLIP-based text prompts on model performance, we extend the original prompt (i.e., $a\ photo\ of\ a/an\ \{\}$.) by three large language models: GPT-4o. (OpenAI, 2023), ERNIE-Bot (Sun et al., 2021), and DeepSeek (DeepSeek-AI et al., 2024). As shown in Table 6, the experimental results indicate that these prompt variations have minimal effect on model accuracy, demonstrating the robustness and practical effectiveness of the proposed method when incorporating CLIP for probability estimation.

# 4 RELATED WORK

In this section, we introduce existing research on learning with privacy concerns (Sportisse et al., 2023), which can be broadly categorized into two main approaches. The first method focuses on aggregate observations (Dan et al., 2021; Roth et al., 2021), where aggregate labels are provided to protect privacy. The second method focuses on inaccurate labeling (Tian et al., 2024; Xu et al., 2023), where each individual sample is labeled with an inaccurate label.

## 4.1 LEARNING FROM AGGREGATE OBSERVATIONS

Aggregate observations are a widely used approach to protect sensitive labels in privacy label learning. Existing methods can be broadly categorized into three classes: pairwise observations (Ramírez et al., 2023; Xia et al., 2024; Liao et al., 2023; Zhang et al., 2020), multiple-instance supervision (Li et al., 2025a) and label proportion learning (He et al., 2023; Asanomi et al., 2023; Liu et al., 2023).

Pairwise observations aim to protect sensitive labels by aggregating data points based on their pairwise relations. Specifically, pairwise comparison (Feng et al., 2021) through comparison of relationships between two instances, while triplet comparison (Cui et al., 2020) compare the relative distance among triplets. Pairwise similarity (Bao et al., 2018) determines whether two instances belong to the same class, and similarity confidence (Cao et al., 2021) further incorporates confidence scores. In addition, ordinal ranking specifies the order between two instances. Multiple-instance supervision focuses on comparing whether at least one positive label exits in an aggregate observations, ensuring that sensitive label information is not directly exposed. Finally, label proportion learning protects sensitive information by providing proportions of data from each class in the aggregate observations, rather than labeling each sample individually.

## 4.2 LEARNING FROM INACCURATE LABELING

Inaccurate labeling methods provide another approach to privacy label learning, where individual sample are labeled, but the labels may be inaccurate to protect sensitive information. Representative approaches include Complementary Label Learning (CoLL) (Lin & Lin, 2023), Partial Label Learning (PLL) (Hai et al., 2025; Huang et al., 2025; Li et al., 2025c; Zhang et al., 2024a), Multi-Positive and Unlabeled Learning (MPUL) (Li et al., 2025b; Perini et al., 2023), and Concealed Labels (CL) (Li et al., 2024), which have been widely studied in recent years.

CoLL supervises the model using labels that indicate classes the instance does not belong to. By learning from these incorrect labels, the method effectively masks the true class of each instance. PLL assigns a set of candidate labels to each instance with only one being correct, thereby avoiding the model using the true label and concealing sensitive information during training. MPUL protects sensitive labels by training models with positive labels and unlabeled instances. In this setting, only positive labels are available, while negative labels remain masked, effectively avoiding exposure of sensitive negative labels. CL aims to protect sensitive information by replacing true labels with either None labels or randomly insensitive labels during training.

Although both aggregate observation and inaccurate labeling methods are effective in protecting sensitive labels, aggregate observation methods often weaken supervision of insensitive labels in order to protect sensitive labels, while inaccurate labeling protects only partial or specific sensitive labels. These limitations highlight the need for approaches that can both protect sensitive labels and maintain full supervision for insensitive labels, as achieved by our proposed augmented supervision.

# 5 CONCLUSION

In this paper, we propose the Aggregate-Masked Labels (AML) to address the issue of protecting sensitive labels during the annotation process. AML maintains partially full supervision for insensitive labels while leveraging aggregate-masked labels to protect sensitive information. Moreover, we introduce a consistent-risk estimator to distinguish between sensitive and insensitive labels, and introduce stochastic sampling with probability estimation to enhance model robustness. Experimental results on both synthetic and real-world datasets demonstrate that AML achieves competitive performance.

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

## A  THE USE OF LARGE LANGUAGE MODELS (LLMS)

During the preparation of this manuscript, large language models (e.g., ChatGPT) were used solely for grammar checking, language polishing and enhancing readability. All initial drafts of the manuscript were written entirely by the authors. The authors carefully reviewed all AI-generated suggestions to ensure accuracy and academic rigor.

## B  PROOFS

### B.1  PROOF OF LEMMA 2

**Lemma 2.** (AMO risk) Under the AML assumption, for multi-class classifier $f$, we derive the expected risk for aggregate observations in the AMO (i.e., $Y_A \in \mathcal{Y}_M$) as follows:

$$
\begin{aligned}
R_{AMO}(f) =& \mathbb{E}_{X \sim p(X)} \sum_{Y_A \in \mathcal{Y}_M} \sum_{Y \in \mathcal{Y}^m} p(Y, Y_A \mid X) \sum_{i=1}^{m} \mathcal{L}\left[f(x_i), y_i\right] \\
=& \mathbb{E}_{(X,Y_A) \sim p(X,Y_A \in \mathcal{Y}_M)} \left[ \sum_{k_1 \in \mathcal{Y}_U \bigcup Y_A} \cdots \sum_{k_m \in \mathcal{Y}_U \bigcup Y_A} \hat{L}(X,K) \right. \\
& \left. - \sum_{k_1 \in \mathcal{Y}_U} \cdots \sum_{k_m \in \mathcal{Y}_U} \hat{L}(X,K) \right]
\end{aligned} \tag{9}
$$

*Proof.* The expected risk for aggregate observations can be expressed as follows:

$$
\begin{aligned}
R(f) &= \mathbb{E}_{(X,Y) \sim p(X,Y)} \sum_{i=1}^{m} \mathcal{L}\left[f(x_i), y_i\right] \\
&= \mathbb{E}_{X \sim p(X)} \sum_{Y \in \mathcal{Y}^m} p(Y \mid X) \sum_{i=1}^{m} \mathcal{L}\left[f(x_i), y_i\right] \\
&= \mathbb{E}_{X \sim p(X)} \sum_{Y_A \subseteq \mathcal{Y}_U \bigcup \mathcal{Y}_M} \sum_{Y \in \mathcal{Y}^m} p(Y, Y_A \mid X) \sum_{i=1}^{m} \mathcal{L}\left[f(x_i), y_i\right] \\
&= \mathbb{E}_{X \sim p(X)} \left[ \sum_{Y_A \in \mathcal{Y}_U} \sum_{Y \in \mathcal{Y}^m} p(Y, Y_A \mid X) \sum_{i=1}^{m} \mathcal{L}\left[f(x_i), y_i\right] \right. \\
&\left. + \sum_{Y_A \in \mathcal{Y}_M} \sum_{Y \in \mathcal{Y}^m} p(Y, Y_A \mid X) \sum_{i=1}^{m} \mathcal{L}\left[f(x_i), y_i\right] \right]
\end{aligned} \tag{10}
$$

On the other hands,

$$
\begin{aligned}
& \mathbb{E}_{X \sim p(X)} \sum_{Y_A \in \mathcal{Y}_M} \sum_{Y \in \mathcal{Y}^m} p(Y, Y_A \mid X) \sum_{i=1}^{m} \mathcal{L}\left[f(x_i), y_i\right] \\
&= \mathbb{E}_{X \sim p(X)} \sum_{Y_A \in \mathcal{Y}_M} p(Y_A \mid X) \sum_{Y \in \mathcal{Y}^m} p(Y \mid Y_A, X) \sum_{i=1}^{m} \mathcal{L}\left[f(x_i), y_i\right] \\
&= \mathbb{E}_{(X,Y_A) \sim p(X,Y_A \in \mathcal{Y}_M)} \sum_{Y \in \mathcal{Y}^m} p(Y \mid Y_A, X) \sum_{i=1}^{m} \mathcal{L}\left[f(x_i), y_i\right] \\
&= \mathbb{E}_{X,Y_A \sim p(X,Y_A \in \mathcal{Y}_M)} \left[ \sum_{k_1 \in \mathcal{Y}_U \bigcup Y_A} \cdots \sum_{k_m \in \mathcal{Y}_U \bigcup Y_A} \prod_{j=1}^{m} P(y_j = k_j \mid x_j) \sum_{i=1}^{m} \mathcal{L}\left[f(x_i), k_i\right] \right. \\
&\left. - \sum_{k_1 \in \mathcal{Y}_U} \cdots \sum_{k_m \in \mathcal{Y}_U} \prod_{j=1}^{m} P(y_j = k_j \mid x_j) \sum_{i=1}^{m} \mathcal{L}\left[f(x_i), k_i\right] \right]
\end{aligned} \tag{11}
$$

Let

$$\hat{L}(X, K) = \prod_{j=1}^{m} P(y_j = k_j \mid x_j) \sum_{i=1}^{m} \mathcal{L}\left[f(x_i), k_i\right] \tag{12}$$

Hence, by substituting eq.(12) into eq.(11)we can obtain

$$\mathbb{E}_{X \sim p(X)} \sum_{Y_A \in \mathcal{Y}_M} \sum_{Y \in \mathcal{Y}^m} p(Y, Y_A \mid X) \sum_{i=1}^{m} \mathcal{L}\left[f(x_i), y_i\right]$$

$$= \mathbb{E}_{(X,Y_A) \sim p(X,Y_A \in \mathcal{Y}_M)} \left[ \sum_{k_1 \in \mathcal{Y}_U \bigcup Y_A} \cdots \sum_{k_m \in \mathcal{Y}_U \bigcup Y_A} \hat{L}(X, K) \right. \tag{13}$$

$$\left. - \sum_{k_1 \in \mathcal{Y}_U} \cdots \sum_{k_m \in \mathcal{Y}_U} \hat{L}(X, K) \right]$$

which proves Lemma 2. □

### B.2 PROOF OF THEOREM 3

**Theorem 3.** Under the AML assumption, the classification risk $R(f)$ can be equivalently expressed as follows:

$$R_{AML}(f) = \mathbb{E}_{X,Y_A \sim p(X,Y_A \in \mathcal{Y}_U)} \sum_{i=1}^{m} \mathcal{L}\left[f(x_i), Y_{A_i}\right]$$

$$+ \mathbb{E}_{X,Y_A \sim p(X,Y_A \in \mathcal{Y}_M)} \left[ \sum_{k_1 \in \mathcal{Y}_U \bigcup Y_A} \cdots \sum_{k_m \in \mathcal{Y}_U \bigcup Y_A} \hat{L}(X, K) \right.$$

$$\left. - \sum_{k_1 \in \mathcal{Y}_U} \cdots \sum_{k_m \in \mathcal{Y}_U} \hat{L}(X, K) \right] \tag{14}$$

*Proof.* The expected risk for aggregate observations can be expressed as follows:

$$R_{AML}(f) = \mathbb{E}_{(X,Y) \sim p(X,Y)} \sum_{i=1}^{m} \mathcal{L}\left[f(x_i), y_i\right]$$

$$= \mathbb{E}_{X \sim p(X)} \sum_{Y \in \mathcal{Y}^m} p(Y \mid X) \sum_{i=1}^{m} \mathcal{L}\left[f(x_i), y_i\right]$$

$$= \mathbb{E}_{X \sim p(X)} \sum_{Y_A} \sum_{Y \in \mathcal{Y}^m} p(Y, Y_A \mid X) \sum_{i=1}^{m} \mathcal{L}\left[f(x_i), y_i\right] \tag{15}$$

$$= \mathbb{E}_{X \sim p(X)} \left[ \sum_{Y_A \in \mathcal{Y}_U} \sum_{Y \in \mathcal{Y}^m} p(Y, Y_A \mid X) \sum_{i=1}^{m} \mathcal{L}\left[f(x_i), y_i\right] \right.$$

$$\left. + \sum_{Y_A \in \mathcal{Y}_M} \sum_{Y \in \mathcal{Y}^m} p(Y, Y_A \mid X) \sum_{i=1}^{m} \mathcal{L}\left[f(x_i), y_i\right] \right]$$

According to Assumption 1, we can obtain

$$\mathbb{E}_{X \sim p(X)} \sum_{Y_A \in \mathcal{Y}_U} \sum_{Y \in \mathcal{Y}^m} p(Y, Y_A \mid X) \sum_{i=1}^{m} \mathcal{L}\left[f(x_i), y_i\right]$$

$$= \mathbb{E}_{X \sim p(X)} \sum_{Y_A \in \mathcal{Y}_U} p(Y_A \mid X) \sum_{i=1}^{m} \mathcal{L}\left[f(x_i), Y_{A_i}\right] \tag{16}$$

$$= \mathbb{E}_{X,Y_A \sim p(X,Y_A \in \mathcal{Y}_U)} \sum_{i=1}^{m} \mathcal{L}\left[f(x_i), Y_{A_i}\right]$$

Hence, from Eq. (15), Eq. (16) and Lemma 2, we have

$$R_{AML}(f) = \mathbb{E}_{(X,Y)\sim p(X,Y)} \sum_{i=1}^{m} \mathcal{L}\left[f(x_i), y_i\right]$$

$$= \mathbb{E}_{X\sim p(X)}\left[\sum_{Y_A\in\mathcal{Y}_U}\sum_{Y\in\mathcal{Y}^m} p(Y, Y_A \mid X)\sum_{i=1}^{m}\mathcal{L}\left[f(x_i), y_i\right]\right.$$

$$\left. + \sum_{Y_A\in\mathcal{Y}_M}\sum_{Y\in\mathcal{Y}^m} p(Y, Y_A \mid X)\sum_{i=1}^{m}\mathcal{L}\left[f(x_i), y_i\right]\right]$$

$$= \mathbb{E}_{X,Y_A\sim p(X,Y_A\in\mathcal{Y}_U)} \sum_{i=1}^{m}\mathcal{L}\left[f(x_i), Y_{A_i}\right]$$

$$+ \mathbb{E}_{X,Y_A\sim p(X,Y_A\in\mathcal{Y}_M)}\left[\sum_{k_1\in\mathcal{Y}_U\bigcup Y_A}\cdots\sum_{k_m\in\mathcal{Y}_U\bigcup Y_A}\hat{L}(X,K)\right.$$

$$\left. - \sum_{k_1\in\mathcal{Y}_U}\cdots\sum_{k_m\in\mathcal{Y}_U}\hat{L}(X,K)\right]$$

which proves Theorem 3. □

## C DETAILS OF DATASETS

The detailed specifications of the datasets are provided in Table 7.

CIFAR-10 and CIFAR-100 consist of natural images with 10 and 100 classes, respectively. Caltech-101 is a standard benchmark dataset for object recognition, containing images from 101 object classes and one background class, totaling 9146 images. DTD (Describable Textures Dataset) consists of 5640 images categorized into 47 texture classes.

AMLM consists of 804 images categorized into six classes. The AMLS contains 925 images from eight classes. Given the sensitive nature of real-world information, certain labels need to be protected during annotation. These datasets were constructed from real-world data related to sensitive medical image, making them highly relevant for evaluating privacy labels learning methods such as AML.

Table 7: Overview of the datasets used in our experiments, including name of dataset, number of Training, number of Testing, and number of Classes.

| Name | Training | Testing | Classes |
|---|---|---|---|
| CIFAR-10 | 50K | 10K | 10 |
| CIFAR-100 | 50K | 10K | 100 |
| Caltech-101 | 6400 | 2746 | 101+1 |
| DTD | 4512 | 1128 | 47 |
| AMLS | 740 | 185 | 8 |
| AMLM | 646 | 158 | 6 |

