# OpenReview forum: "Learning from Aggregate-Masked Labels"
_ICLR.cc/2026/Conference — Submitted to ICLR 2026_

### Official Review · Reviewer_t64X · 2025-10-27

**Soundness:** 2
**Presentation:** 2
**Contribution:** 2
**Rating:** 4
**Confidence:** 3

**Summary:**

This work studies how to train classifiers using aggregate observations so as not to compromise data privacy. The authors propose Aggregate-Masked Labels (AML), a setting that augments aggregate supervision with full labels for non-sensitive attributes. They derive a risk-consistent estimator tailored to AML and introduce a stochastic label combination method to lower training overhead. Experimental results highlight the effectiveness of proposed methods.

**Strengths:**

1. The paper explores an interesting and practically relevant setting that seeks a more fine-grained trade-off between privacy and performance, rather than adopting weak or full supervision in an all-or-nothing manner.
2. Comprehensive experiments on both benchmark and real-world datasets demonstrate the superiority and robustness of the proposed method.

**Weaknesses:**

1. Potential issue in Eq. (7). My understanding is that the authors first enumerate all possible combinations and then subtract the combinations that contain only insensitive labels. However, in the current Eq. (7), after subtracting the “all-insensitive” cases, what remains are combinations that contain at least one sensitive label.
If my understanding is correct, we should retain only combinations in which the aggregate set includes all sensitive labels in $\mathcal{Y}_M$ (rather than only a subset of them). This distinction becomes problematic when $|\mathcal{Y}_M| > 1$.

2. Lack of experimental validation. The authors state that full supervision is introduced to enhance model performance, yet this claim lacks empirical support. The manuscript does not report how much improvement is achieved after adding full supervision.

3. Missing citations. There are several missing citations throughout the paper. I recommend that the authors carefully check and supplement the citations.
e.g.

  ·  Line 253: The CLIP model should be properly cited.

  ·  Section 4.1: There are missing references for ordinal ranking, multiple-instance learning, and learning from label proportions.

**Questions:**

In Tables 1, 3, and 6, the titles of the last two columns are both written as {R, R}. Could the authors clarify what the difference is between these two columns?

---

### Official Review · Reviewer_fan5 · 2025-10-31

**Soundness:** 2
**Presentation:** 2
**Contribution:** 1
**Rating:** 2
**Confidence:** 2

**Summary:**

This paper proposes a privacy-preserving machine learning algorithm to protect sensitive labels. The idea is to aggregate training data into small groups, and use masked label if the group contains sensitive labels. If a group only contains insensitive labels, then the original raw labels will be used. Risk bound is derived for this novel setting. Experiments are conducted to show that the proposed method can beat existing methods including Learning from Partial Labels and Learning from Label Proportion.

**Strengths:**

- The idea of learning from aggregate-masked labels is interesting.
- The experiment results are extensive and supportive.

**Weaknesses:**

- Although privacy seems to be the motivation of this work, this paper relies entirely on an ad-hoc definition of privacy ("masking," "aggregate observations") rather than a formal privacy guarantee like Differential Privacy (DP) or k-anonymity.
  - The term "protecting sensitive labels" is used frequently, but without any quantitative metric or formal proof, there is no way to quantify how much privacy is actually achieved or against what type of adversary the method is secure.
  - Experiments are conducted to compare the accuracy of the proposed method with multiple existing methods. However, since none of the methods are placed within a common, formal privacy framework (like being guaranteed to the same $\epsilon$ level), there is no way to know if they are at the same privacy level. Therefore, claiming that the larger accuracy of the proposed method implies it is a better algorithm is misleading; it may simply be less private. The paper only demonstrates a utility gain, not a favorable utility-privacy trade-off.
- I also do not get the threat model. Who is the adversary, and what is exposed to the adversary? According to the paper, it seems that there are two parties, one for data annotation and the other for model training. The annotation party has access to the raw data. It can operate on the data and pass the filtered/processed data to the model training party. The goal seems to be that the party for model training cannot see sensitive labels. Can you confirm if this is the set up? If so, can you elaborate a bit more on the practical scenarios of such settings?

**Questions:**

- The tables are not very clear. For instance, there are two columns with index ${R,R}$. What does this mean, and why we have two columns for this?

---

### Official Review · Reviewer_xQs6 · 2025-10-31

**Soundness:** 3
**Presentation:** 2
**Contribution:** 3
**Rating:** 4
**Confidence:** 3

**Summary:**

This paper introduces the Aggregate-Masked Labels (AML) framework to address the privacy risk of exposing sensitive labels in supervised learning. The method allows full label supervision for insensitive labels while masking only the sensitive components within aggregations, thus retaining strong supervision where possible and providing privacy protection.

**Strengths:**

A formal problem statement is provided, with explicit assumption definitions, and mathematical derivations.

Strong Empirical Results

**Weaknesses:**

The mathematical presentation in Section 2 and its equations is dense; informal explanation or worked examples are missing.

Unclear definition of sensitive labels and realism. In benchmarks (CIFAR, Caltech, DTD), the notion of “sensitive” seems arbitrary (random selection of labels). It is unclear whether such synthetic settings meaningfully reflect privacy constraints or affect supervision distribution.

The model uses CLIP as the backbone, which largely dominates the performance. However, there is no ablation study to demonstrate that the improvement actually comes from the proposed method rather than the powerful CLIP backbone itself.

**Questions:**

See above.

---

### Official Review · Reviewer_eNdD · 2025-11-02

**Soundness:** 3
**Presentation:** 3
**Contribution:** 2
**Rating:** 6
**Confidence:** 3

**Summary:**

This paper introduces the Aggregate-Masked Labels (AML) setting, which aims to protect sensitive labels during data annotation while still maintaining partial full supervision. When an aggregate contains sensitive labels, only the set of sensitive labels is revealed, and ground truth labels are masked. When no sensitive labels appear, true labels for all instances are provided. To learn under this setting, the authors propose a risk-consistent estimator that enables effective training without exposing sensitive labels, and a stochastic label combination strategy to reduce computational cost. Experiments across several benchmark and real-world datasets show clear improvements over existing weakly supervised and privacy-preserving methods.

**Strengths:**

1. The paper proposes a novel privacy-aware learning setting that protects sensitive labels while keeping supervision for insensitive labels, together with a statistically consistent learning method. This makes the formulation practically useful and theoretically sound.

2. The proposed stochastic label combination greatly improves computational efficiency, making the method scalable to larger aggregate sizes.

3. Extensive experiments on multiple datasets, including real-world cases, demonstrate strong performance and practical relevance.

**Weaknesses:**

1. The connection to multi-instance learning (MIL) should be discussed. AML resembles MIL, since both provide supervision at the group level rather than per instance. For AMO, revealing only whether sensitive labels exist is conceptually similar to the MIL assumption that a bag is positive if it contains at least one positive instance. However, the paper does not clarify how AML fundamentally differs from MIL or why existing MIL techniques are insufficient. A brief comparison would help position the contribution more clearly.

2. The method heavily relies on probability estimation quality, yet the impact of inaccurate probability estimates is not analyzed. A robustness discussion or theoretical bound would strengthen the work.

3. Since AML also works with grouped samples, a short mention of Label Proportion Learning [1] would help readers see how your setting compares with existing group-based supervision. Also, Complementary Label Learning [2] is often motivated by privacy concerns, so briefly acknowledging its relevance would make the literature positioning feel more complete.

[1]. Jianxin Zhang, Yutong Wang, and Clayton Scott. Learning from Label Proportions by Learning with Label Noise. NeurIPS 2022.

[2]. Shuqi Liu, Yuzhou Cao, Qiaozhen Zhang, Lei Feng, and Bo An. Consistent Complementary-Label Learning via Order-Preserving Losses. AISTATS 2023.

**Questions:**

Please see the weaknesses.

---

### Meta-Review · Area_Chair_ACQT · 2025-12-23

**Summary:**

Some reviewers agree the paper proposes an interesting privacy-motivated supervision setting (AML) with risk-consistent training and shows strong empirical performance. However, reject recommendation is driven by several high-impact concerns:

A central reviewer concern is that the paper claims “protecting sensitive labels” but does not define a clear threat model nor provide a quantitative/privacy guarantee. Without a measurable privacy objective, it is hard to assess whether gains come from a favorable utility–privacy trade-off or simply weaker privacy.

Multiple reviews note missing or insufficient discussion of how AML differs from or subsumes related settings such as MIL, label proportion learning, and complementary label learning, which weakens the novelty/claims and makes it difficult to interpret the contribution boundary.

The presentation is dense with missing informal explanations/examples, and one reviewer raises a possible issue in a key equation that could affect correctness when aggregates contain multiple sensitive labels.

The use of a strong CLIP backbone without a convincing ablation raises the possibility that reported improvements are dominated by the backbone rather than the AML-specific estimator/strategy. In addition, the claimed benefit of adding full supervision for non-sensitive labels is not cleanly isolated.

**Reviewer Concerns:**

Concerns likely addressed:

These are straightforward to fix in revision, and likely addressed if authors were attentive.

Table notation confusion: Also likely fixable with clarification.

Concerns that are still outstanding:

This is the most critical issue. If the paper cannot specify (i) who the adversary is, (ii) what information is exposed, and (iii) what privacy criterion is guaranteed or measured, then the central privacy claim remains ungrounded. Even if the approach is practically motivated, the paper needs at least a well-defined leakage metric or a formal notion.

If “sensitive” is randomly chosen in standard datasets, the setting may not reflect realistic privacy constraints; without a principled sensitivity model or real-world justification, conclusions about privacy-preserving learning are weakened.

If this concern is valid, it directly affects soundness. A rebuttal would need a precise derivation and/or correction, plus a check that experiments remain valid.

Adding citations alone is not enough; the paper should clearly articulate how AML differs, when it reduces to known settings, and why existing techniques are insufficient.

Without controlled ablations, it remains unclear whether the proposed AML-specific innovations drive the gains.

**Reviewer Scores:**

fan5: 2–>3

t64X: 4->4

xQs6: 4->4

eNdD: 6->6

---

### Decision · Program_Chairs · 2026-01-26

Reject